# Mitigating Bias in Scientific Data:
# A Materials Science Case Study

**Hengrui Zhang**
Northwestern University
hrzhang@u.northwestern.edu

**Wei (Wayne) Chen**[*]
Texas A&M University
w.chen@tamu.edu

**James M. Rondinelli**[†]
Northwestern University
jrondinelli@northwestern.edu

**Wei Chen**[†]
Northwestern University
weichen@northwestern.edu

## Abstract

Growing scientific data and data-driven informatics drastically promote scientific discovery. While there are significant advancements in data-driven models, the quality of data resources is less studied despite its huge impact on model performance. As an example, we focus on data bias arising from uneven coverage of materials families in existing knowledge. Observing different diversities among crystal systems in common materials databases, we propose an information entropy-based metric for measuring this bias. To mitigate the bias, we develop an entropy-targeted active learning (ET-AL) framework, which guides the acquisition of new data to improve the diversity of underrepresented crystal systems. We demonstrate the capability of ET-AL for bias mitigation and the resulting improvement in downstream machine learning models. This approach is broadly applicable to data-driven materials discovery, including autonomous data acquisition and dataset trimming to reduce bias, as well as data-driven informatics in other scientific domains.

## 1 Introduction

Data-driven research has emerged as a new paradigm for scientific discovery [1–3]. Take materials science as an example: with large materials data and powerful informatics tools, this paradigm significantly accelerates the understanding, predictive modeling, and design of materials [4–7]. While the informatics tools, such as machine learning (ML) models, hold a conspicuous position in these works, the data resources are as important [8, 9]. The performances that the models can attain highly depend on the quality of data they are built upon. Data veracity entails a description of where and how data were collected, but less frequently is why (or why not) using certain data clearly articulated.

Following the Materials Genome Initiative, multiple materials data resources [10–12] have been constructed using high-throughput first-principles calculations. Besides these centralized data resources, a growing portion of materials data is generated in various research projects, available from published papers and repositories, and increasingly utilized owing to data/text mining tools [13]. However, it is common that materials data do not have uniform coverage:

1. The candidate materials for database construction are selected among known structures or based on known structural prototypes, and lower symmetry structures are less explored than higher symmetry ones.

---

[*]Work performed while at Northwestern University.
[†]Corresponding authors.

NeurIPS 2023 AI for Science Workshop.

2. Most literature only reports compounds perceived to exhibit "good" properties based on the aspect of interest, while the "unsatisfactory" results can also be valuable [14].

3. Property simulation is easier for compounds that are structurally simple, and property measurement is simpler for compounds that are readily synthesizable and stable at ambient pressures and temperatures.

These, among other factors, lead to bias in the materials data platforms.

Data bias, a ubiquitous issue in data science, has been more recognized in the social science domain [15, 16] but is often overlooked in physical sciences, including materials science. Just as it causes social inequity in social policy built upon that data, bias in materials data is harmful to data-driven materials modeling and design [17]. An example is a bias in stability data among crystal structures, which we refer to as "structure–stability bias". Such bias hinders the modeling of microstructure, thus affecting the accurate prediction of various materials properties [18].

Although some attempts have been pursued to characterize bias on trained models *post facto* [19] or reduce the impact of data bias on model training [20], few have addressed bias intrinsic to the data for which the models are trained and mitigated bias from the onset. As bias originates from uneven coverage of different materials families, it can be captured by examining the diversities of families in the data, which reflects the completeness of coverage. Moreover, by adding well-selected new data points, bias in a dataset can be reduced. Towards this end, the active learning (AL) method provides a way to sequentially select optimal data points guided by sampling strategies considering uncertainty, diversity, or performance [21–24]. With a specially designed sampling strategy, AL can serve as a method for bias reduction. To this end, we propose an entropy-targeted active learning (ET-AL) algorithm as a systematic approach to bias reduction [25]. We show that ET-AL provides a general method for mitigating bias in materials datasets and is also applicable in guiding the construction of materials databases, thus granting materials researchers access to low-bias data for machine learning.

## 2 Data Bias Characterization

For demonstration purposes, we retrieve two materials datasets:

1. Structure and formation energy per atom of all binary intermetallic compounds among the elements Al, Ti, Cr, Fe, Co, Ni, Cu, and W from the Open Quantum Materials Database (OQMD) [11] (denoted `OQMD-8`, size 2953).

2. All entries with elastic moduli available from the JARVIS classical force-field inspired descriptors (CFID) dataset [26], excluding elements H, Tc, halogens, noble gases, lanthanum family, and those with atomic numbers $\geq 84$ (denoted `J-CFID`, size 10898).

We show the distribution of formation energy per atom $\Delta E$ of materials in the two datasets with respect to crystal system in Figure 1(a–b). Among the seven crystal systems, the lower symmetry monoclinic and triclinic systems display higher distribution density in the more stable (lower $\Delta E$) region. This observation contradicts the empirical rules that materials with higher symmetry (which are usually more close-packed and have higher coordination numbers) generally have higher stability [27]. Such contradiction is due to the imbalanced coverage of different crystal systems in the materials datasets, and we refer to this problem as "structure–stability bias".

Without assuming any prior knowledge such as the correlation between symmetry and stability, are we still able to capture the bias? Data bias can be characterized as "unjustifiable concentration on a particular part" [28]. In the context of materials science, the "parts" can be regions in a space broadly by composition, (micro)structure, property, processing, or energy-based descriptions. As an example, the structure–stability bias arises from concentration (uneven coverage) in materials with certain structures (crystal systems) and stability ($\Delta E$). Such uneven coverage can be captured by the different diversities of stability among different crystal systems. To quantify that, we first define the diversity of a dataset by recognizing that for values of a continuous variable $Y$, the diversity can be quantified by information entropy

$$h(Y) = \mathbb{E}[-\log f(Y)] = -\int f(y) \log(y) \mathrm{d}y, \tag{1}$$

where $f(y)$ is the underlying probability density function of $Y$. For a finite set of $Y$ value, $h(Y)$ can be estimated numerically [29]. In general, we can group the data into clusters by any appropriate

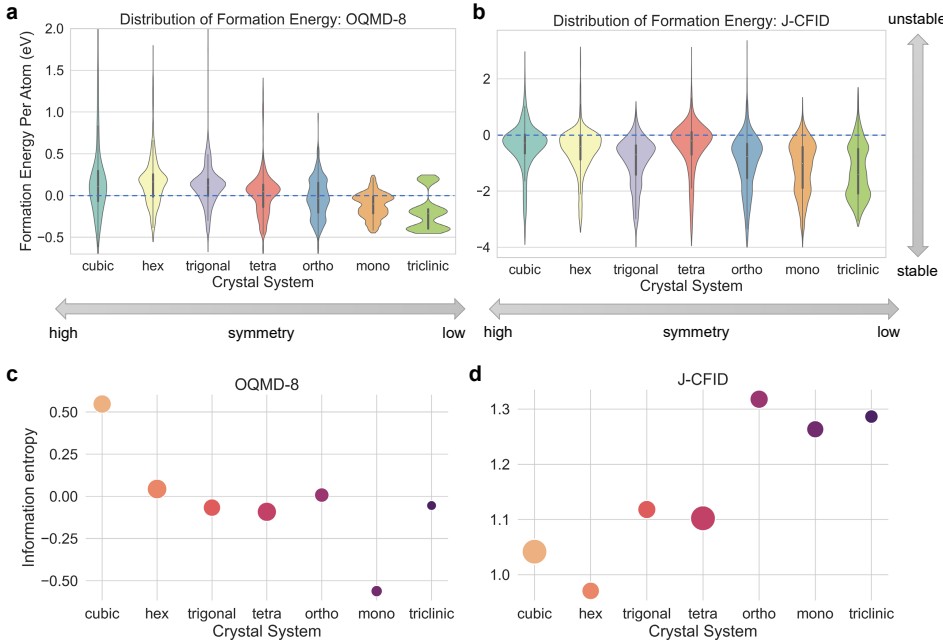

Figure 1: **Structure–stability bias in two datasets.** (a–b) Kernel density estimation of the distribution of formation energy among different crystal systems in the `OQMD-8` and `J-CFID` datasets. (c–d) Information entropy of formation energy among different crystal systems in `OQMD-8` and `J-CFID` datasets. Colors indicate the degrees of symmetry of crystal systems; the sizes of points reflect the number of datapoints in each crystal system. Note that the `OQMD-8` dataset contains only 5 triclinic materials, which causes inaccuracy in the information entropy estimation for the triclinic system.

criterion and estimate $h(Y)$ for every cluster from the $Y$ values in the dataset, thus quantifying the diversity of $Y$ in every cluster. Based on the observation from Figure 1(a–b), the `OQMD-8` dataset has coverage of materials with diverse $\Delta E$ values in the high symmetry crystal systems, while the $\Delta E$ values in the triclinic and monoclinic systems are not diverse. The `J-CFID` dataset, on the other hand, lacks diversity in the high-symmetry crystal systems.

Next, we measure bias using a fairness criterion, i.e., the difference in $h(Y)$ between different clusters indicate the existence and level of bias. For our application, we will use crystal systems as natural clusters, and quantify the structure–stability bias via fairness of $h(\Delta E)$. Figure 1(c–d) shows that $h(\Delta E)$ captures the observed difference in diversities, thus reflecting the structure–stability bias.

## 3 Active Learning for Bias Mitigation

With fairness in diversity as a measure, the data bias can be reduced systematically by adding data to the least diverse group in a manner that increases its $h(Y)$. We develop the entropy-targeted active learning method (Figure 2) to attain this automatically. In the active learning context, we refer to the materials with properties known and unknown as "labeled" and "unlabeled", respectively. The ET-AL algorithm iteratively picks a target group (the least diverse one with unlabeled samples available), selects an optimal unlabeled material that may improve $h(Y)$ of the group, and adds it to the labeled data. The iteration terminates when a pre-specified criterion is satisfied, or all available samples are labeled.

When a target group is selected, a Gaussian Process (GP) model [30] is fitted to the labeled data of that group, with any type of structure representation as predictors and $Y$ as the response. By assuming that the responses $\boldsymbol{y} = \{y_1, \ldots, y_m\}$ are jointly Gaussian distributed with noise given the predictors $\boldsymbol{X} = \{\boldsymbol{x}_1, \ldots, \boldsymbol{x}_m\}$:

$$\boldsymbol{y}|\boldsymbol{X} \sim \mathcal{N}\left(\boldsymbol{\mu}, \boldsymbol{K} + \sigma^2 \boldsymbol{I}\right), \qquad (2)$$

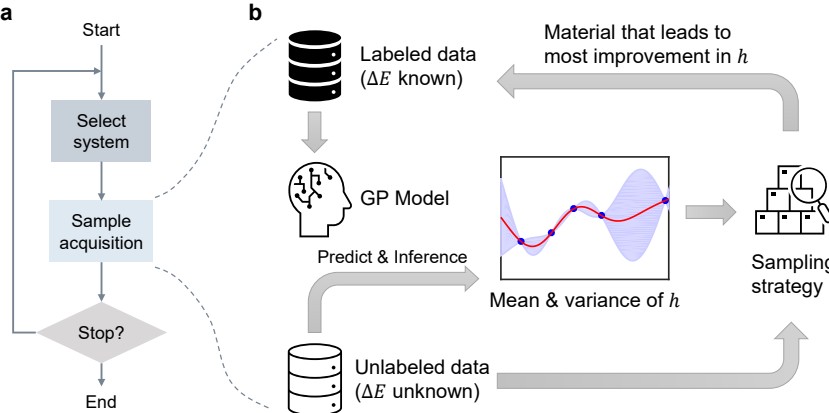

Figure 2: **Schematic of the ET-AL algorithm for data bias mitigation.** (a) Overall procedure of ET-AL: a target crystal system is selected, then an unlabeled material is selected and labeled. The steps repeat until the stopping criteria are satisfied. (b) The procedure of sample acquisition: a Gaussian Process (GP) model is trained with the labeled data and makes predictions for the unlabeled data. The predictive mean and variance of $h$ resulting from adding each material are inferred therefrom. Based on these, the optimal material is selected according to the sampling strategy and added to the labeled data.

GP makes uncertainty-aware predictions, i.e., not a single value but a distribution of the response. Leveraging this, for each unlabeled sample in the target group, the GP model provides a predicted distribution of $Y$, from which we use the Monte Carlo method to draw samples and infer the resulting change in $h(Y)$ by adding the sample. We thereby obtain the predictive mean and variance of $h$ for every unlabeled sample, which are subsequently used in the evaluation of an acquisition function, such as expected improvement (EI) [31]:

$$\text{EI}(\boldsymbol{x}) = \mathbb{E}\left[\max\{0, \Delta(\boldsymbol{x})\}\right] = \hat{s}(\boldsymbol{x})\phi\left(\frac{\Delta(\boldsymbol{x})}{\hat{s}(\boldsymbol{x})}\right) + \Delta(\boldsymbol{x})\Phi\left(\frac{\Delta(\boldsymbol{x})}{\hat{s}(\boldsymbol{x})}\right), \tag{3}$$

where $\Delta(\boldsymbol{x})$ is the difference between the predicted mean $h$ and the current $h$; $\hat{s}(\boldsymbol{x})$ is the predicted standard deviation of $h$; $\phi(\cdot)$ and $\Phi(\cdot)$ are the standard Gaussian distribution functions. The unlabeled sample with the largest EI is selected. This completes an iteration of ET-AL. Algorithm 1 provides a formal description of the ET-AL method.

---

**Algorithm 1:** Entropy-targeted active learning.

---

**Data:** Labeled dataset $\mathcal{D} = \{\boldsymbol{x}_i, y_i\}$, unlabeled dataset $\mathcal{U} = \{\boldsymbol{x}'_j\}$; $\mathcal{D} = \cup_c \mathcal{D}_c, \mathcal{U} = \cup_c \mathcal{U}_c$;
      Monte Carlo sample size $n$; stopping criteria
**Result:** Augmented dataset $\mathcal{D}$
**while** *stopping criteria not satisfied* **do**

    Calculate information entropies $H(\mathcal{D}_c) = h(y \in \mathcal{D}_c)$, select $c^* = \arg\min_c H(\mathcal{D}_c)$
    Fit a GP to $\mathcal{D}_{c^*}$: $Y \sim \mathcal{GP}(\boldsymbol{X})$
    **for** $\boldsymbol{x}'_j \in \mathcal{U}_{c^*}$ **do**

        Draw $n$ samples $\left\{y_j^{(k)}\right\}_{k=1}^n$ from $\mathcal{GP}\left(\boldsymbol{x}'_j\right)$
        Calculate $h_j^{(k)} = H\left(\mathcal{D} \cup \left\{y_j^{(k)}\right\}\right)$ **for** $k = 1$ **to** $n$
        Calculate the mean and variance of $\left\{h_j^{(k)}\right\}_{k=1}^n$, EI$\left(\boldsymbol{x}'_j\right)$ according to Equation 3

    **end**
    Select sample $\boldsymbol{x}^* = \arg\max_{\boldsymbol{x}' \in \mathcal{U}_{c^*}} \text{EI}(\boldsymbol{x}')$
    Acquire $y^*$, remove $\boldsymbol{x}^*$ from $\mathcal{U}_{c^*}$, add $\{\boldsymbol{x}^*, y^*\}$ to $\mathcal{D}_{c^*}$
**end**

---

# 4 Experiments

As a demonstration of the ET-AL method, we conduct experiments on the `J-CFID` dataset. The overall procedure is illustrated in Figure 3(a): we split the dataset into a test set (size $N_{\mathrm{T}}$), a labeled set (size $N_{\mathrm{L}}$) with artificial bias, and an unlabeled set (size $N_{\mathrm{U}}$). We use ET-AL to augment the labeled set into a low-bias training set (marked `ETAL`) and create another training set (marked `RAND`) of the same size by randomly sampling from the unlabeled set. In addition to demonstrating that ET-AL effectively reduces the structure–stability bias, we show the impact such bias has by comparing supervised ML models for bulk modulus $B$ and shear modulus $G$ derived from the two training sets.

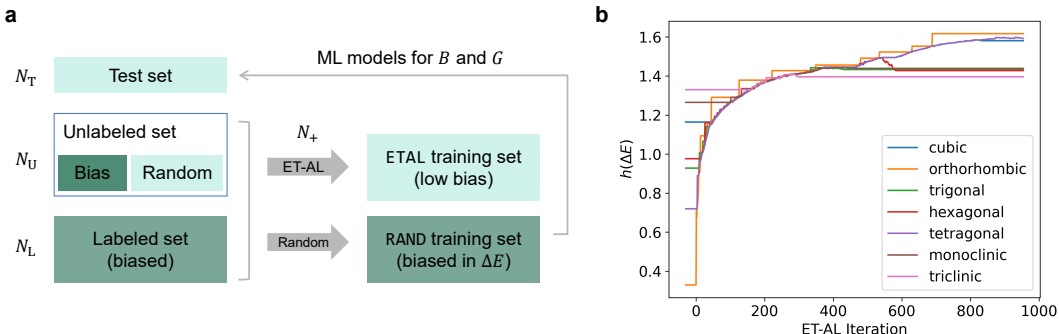

Figure 3: **Experiments on the `J-CFID` dataset.** (a) Split of the dataset: $N_{\mathrm{T}}$ entries are left out as the test set. From the remaining data, some entries are taken away to create an artificial bias and put into the unlabeled set together with randomly selected entries, in total $N_{\mathrm{U}}$. The $N_{\mathrm{L}}$ entries remaining form a labeled set with significant bias. Two training sets are constructed by adding the same number of samples ($N_+$) from the unlabeled set to the labeled set, guided by ET-AL and randomly, respectively. (b) Change of information entropy in every crystal system during ET-AL iterations.

In the experiment, we set $N_{\mathrm{L}} = 1000$, $N_{\mathrm{U}} = 4898$, and $N_{\mathrm{T}} = 5000$. The artificial bias is introduced by removing all tetragonal and trigonal materials with $\Delta E > 0$ and all orthorhombic materials with $\Delta E < 0$. We represent the materials by 32-dimensional graph embedding vectors by feeding their structures to a pre-trained graph neural network model [32] and obtaining the activations of the last but one layer of neurons. ET-AL is applied to the dataset and runs for 954 iterations before termination. As Figure 3(b) shows, the introduced bias is captured by the diversity metric (the three manipulated crystal systems have relatively low initial $h$), and mitigated by ET-AL. Moreover, through ET-AL, the dataset reaches a state where diversities of crystal systems are closer to each other as compared to the initial state, which is favored by the fairness criterion.

Next, we investigate the effects of ET-AL on dataset distribution. We employ t-distributed stochastic neighbor embedding (t-SNE) for dimension reduction of the graph embedding representations of `J-CFID` data into a 2-dimensional space. The low-dimensional embeddings acquired by t-SNE reflect the distribution of data in the structure space. In Figure 4, we use these embeddings to show the coverage of the labeled dataset and the ET-AL-selected and randomly selected data. ET-AL guides sampling in the underrepresented regions (lighter shades in Figure 4 (a), as opposed to a nearly uniform coverage by random sampling in Figure 4(c).

To assess the impact of bias on property prediction, we train multiple supervised learning models on the two training sets, both of size 1954, for predicting $B$ and $G$ from 117 physical descriptors (retrieved using an automatic materials featurization workflow [33]). Each model is trained 30 times with different random states (controlling the initialization, feature permutation, etc., but not affecting training data), with hyperparameters selected using 5-fold cross-validated grid search, and the coefficient of determination ($R^2$) on the test set is recorded. Models include random forest (RF), gradient boosting (GB), shallow neural network, and support vector regression (SVR), among which RF and GB attain relatively better performances on the task. A potential reason for such performance difference is that the descriptors form heterogeneous tabular data, for which tree ensemble models have an advantage [34]. We summarize the performances of these ML models in Figure 5(a), from which we find that models derived from the `ETAL` dataset with reduced bias display systematically superior accuracies over those from the `RAND` dataset.

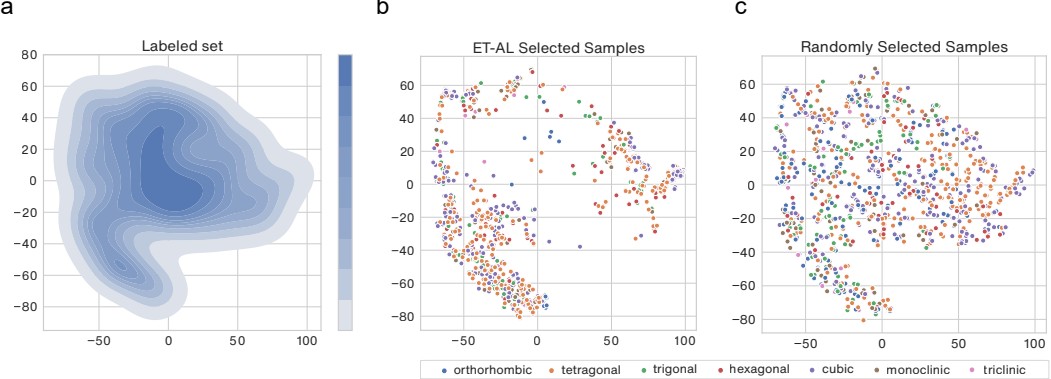

Figure 4: **Visualization of dataset distributions.** (a) Kernel density estimation (KDE) plot of t-SNE embeddings of the labeled dataset. The shade shows the density of points, regions with lighter colors are less covered. (b–c) t-SNE plots of graph embeddings of the materials selected by ET-AL and random sampling, respectively, with colors indicating crystal systems. Compared to random sampling, materials selected by ET-AL better cover the region where labeled samples are sparse, as well as the crystal systems where artificial bias is introduced.

In Figure 5(b), we mark the "most improved samples", i.e., testing samples for which the ML models' prediction accuracies using the `ETAL` training set show greater advantages compared to using the `RAND` training set. As observed, most of these samples are in the underrepresented regions of the labeled set (low-density regions in 4(a)). ET-AL's focus in these regions during sample selection (triangles in Figure 5(b)) overlap with sampling points in Figure 4(b)) leads to the better accuracy of ML models trained on the `ETAL` dataset. These observations agree with the findings of [35]: ML models trained on a biased dataset lack generalizability to underrepresented test samples. ET-AL provides a solution to the problem by reducing structure–stability bias, which improves the coverage of the dataset in the structure space, and thus facilitates downstream tasks such as ML modeling of mechanical properties $B$ and $G$.

# 5 Discussions

## 5.1 Algorithm Complexity

The execution of ET-AL consists of three main parts: (1) GP model fitting on the labeled data, (2) selection of an unlabeled datapoint, and (3) label acquisition. The time part (3) takes depends on the experimental/computational technique used, and it is usually the most time-consuming component, compared to which the time of (1) and (2) are negligible. Quantities affecting the computational time of parts (1) and (2) include the size of the labeled dataset $N_{\mathrm{L}}$, size of the unlabeled dataset $N_{\mathrm{U}}$, and the number of Monte Carlo samples $n_{\mathrm{MC}}$.

For part (1), GP fitting, the time complexity is $\mathcal{O}\left(N_{\mathrm{L}}^3\right)$. The scalable GP models [36] provide a solution for better scalability on large datasets. In particular, the sparse variational (SV) GP model reduces the time complexity to $\mathcal{O}\left(m^3\right)$ (where $m \ll N_{\mathrm{L}}$ is the number of inducing points whose distribution is representative of the training data), with a slight loss of accuracy.

In part (2), for every unlabeled data, the GP model makes one prediction, costing computational time $\mathcal{O}\left(N_{\mathrm{U}}\right)$; then $n_{\mathrm{MC}}$ Monte Carlo samples are drawn, and for each sample the information entropy and acquisition function are calculated, costing computational time $\mathcal{O}\left(N_{\mathrm{U}} \cdot n_{\mathrm{MC}}\right)$. The total time complexity of part (2) is $\mathcal{O}\left(N_{\mathrm{U}}\left(k + n_{\mathrm{MC}}\right)\right)$, where $k$ is constant.

## 5.2 Potential Applications and Outlook

The data bias metric and ET-AL method proposed in this work have a wide range of applications in materials discovery and beyond. First, researchers may examine and potentially reduce the bias in their datasets before developing data-driven models thereon or publishing the data. Second, ET-AL

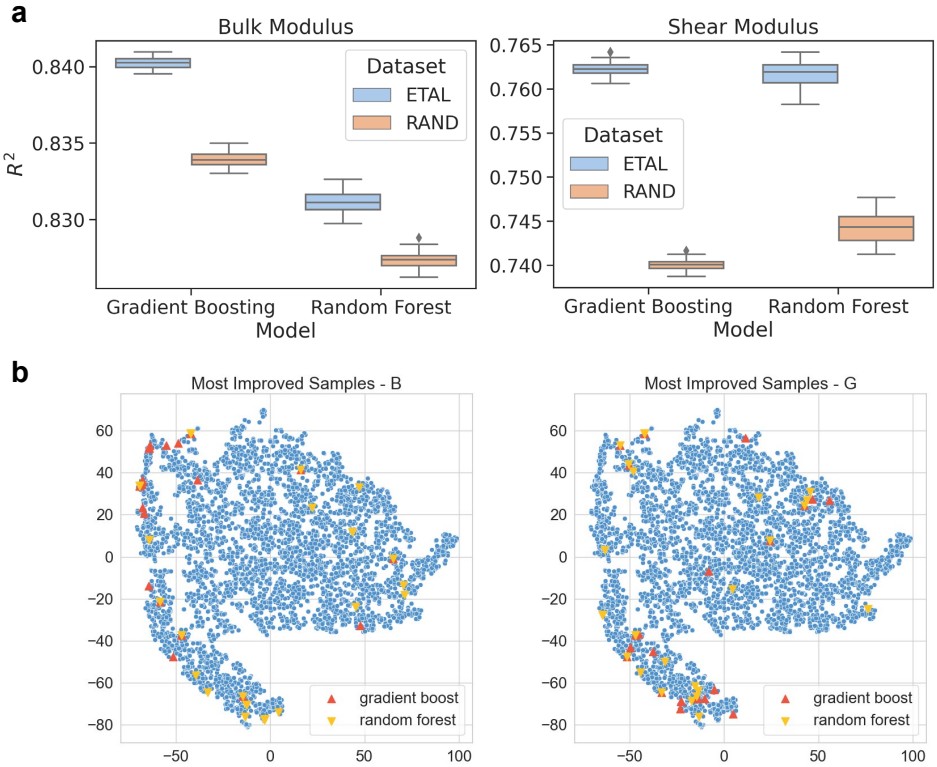

Figure 5: **(a) Comparison of supervised ML models trained on two `J-CFID`-derived datasets.** The boxplots show testing $R^2$ of each model type and target property across 30 replicates, for bulk modulus (left) and shear modulus (right). **(b) Locations of the most improved samples.** Blue dots show the t-SNE mapping of the test samples' graph embeddings. The triangles mark the 20 samples where each ML model trained on the `ETAL` dataset shows the most advantages in accuracy over those trained on the `RAND` set, for bulk modulus (left) and shear modulus (right).

allows steering autonomous data acquisition in an unbiased way. This includes high-throughput computation, as well as experiments such as self-driving laboratories. An application of particular significance is dealing with bias in materials data resources. Since new materials are continually added to the databases, ET-AL can fit in the pipeline to select the materials to add. In practice, however, some databases are so large that an observable effect of bias mitigation requires adding many new data points, and there are other considerations besides bias in database construction. In remediation, the information entropy-based bias metric can also guide trimming rather than expanding a database, i.e., selecting a less biased subset, with the level of bias tunable.

ET-AL is a preliminary step toward mitigating data bias and may be improved in several ways, to name a few: (1) Use a more physically meaningful representation of materials, together with other uncertainty-aware ML models, e.g., Bayesian neural network. (2) Consider multi-fidelity label acquisition ("oracles" in active learning) by incorporating various computational and experimental techniques. (3) Use generative models to acquire new samples independent of a pre-specified sample pool. As pointed out for materials data, bias may exist in other data-intensive research fields, especially where large data are generated and curated for future reuse. Besides offering a solution, we also hope this work can raise the awareness of bias in scientific data that is worth further exploration.

## 6   Conclusion

We highlighted the previously overlooked bias in materials data resources, which has an impact on a broad range of data-driven materials modeling and design studies. We proposed a generic metric for data bias based on diversity measured by information entropy, which successfully captures the

structure–stability bias in datasets retrieved from widely used materials data platforms OQMD and JARVIS. We then formulated and implemented an entropy-target active learning (ET-AL) framework to automatically reduce bias in datasets by acquiring new samples. Through ablation studies, we demonstrated that ET-AL can effectively reduce the structure–stability bias, thus improving data coverage in the structure space and increasing the accuracy of data-driven modeling of materials properties.

We also note that as a generic framework, ET-AL's capability is not limited to materials databases. As the data-driven research paradigm has been adopted by various domains, and data bias is ubiquitous in almost every data system, we anticipate that the ET-AL method is applicable to a variety of scientific and engineering domains, to facilitate the curation of high-quality data and data-driven studies. The code that implements the ET-AL algorithm is available at https://github.com/Henrium/ET-AL.

## Acknowledgments and Disclosure of Funding

Support from the Advanced Research Projects Agency-Energy (DE-AR0001209) and the Center for Hierarchical Materials Design (ChiMaD NIST 70NANB19H005) are greatly appreciated. The authors acknowledge Francesca Tavazza and Brian DeCost for assistance with data collection. H.Z. thanks Kyle Miller, Dale Gaines II, Adetoye Adekoya, Whitney Tso, and Jeffrey Snyder for insightful discussions, and Ke Sun for valuable advice on visualization.

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
