# OpenReview forum: "Mitigating Bias in Scientific Data: A Materials Science Case Study"
_NeurIPS.cc/2023/Workshop/AI4Science — NeurIPS2023-AI4Science Poster_

### Official Review · Reviewer_MCq5 · 2023-10-06
**A new approach to debias the data distribution for crystal systems**

**Rating:** 6
**Confidence:** 4

**Review:**

**Strength**
1. The paper and the storyline are well-written with good motivation.
2. The proposed framework, entropy-targeted active learning (ET-AL), is computationally efficient and easy-to-understand.
3. The sample acquisition with the Gaussian Process (GP) is effective and can adapt to other domains.

**Weakness**
1. It will be beneficial to the framework if authors can show the superiority of the framework compared to adaptive data augmentation approaches (e.g., AutoAugment, CutMix).
2. Instead of relying on GP to obtain the likelihood of future data. using a lightweight generative model (e.g., VAE) can be possibly more effective. Authors should argue the infeasibility of doing so.

Overall, this domain is worthy of further exploration, and I will rate this paper as a borderline acceptance.

---

### Official Review · Reviewer_PD5Q · 2023-10-21
**An intereting paper abut bias-removal learning**

**Rating:** 6
**Confidence:** 4

**Review:**

In this paper, the authors offer an approach to reduce the bais exisiting in different datasets for materials. Their method is based on entropy filtering and active learning. Overall, it is a wel-written paper and the improvement is clear. However, I have some questions about their content:

1. The motivation of using Gaussian Process to fit the labels are not clear. There are also different approaches to model the distribution of Y, for example, varitional inference with output distribution modeling. I am doubting if it is the best approach to model the output distribution.

2. It seems that the authors only compare their method based on GB and RF, but since these two methods are similar, the generalization ability could be challenged. I am more interested in the peformance improvement of such method to neural network based approach. If the improvement is not very significant, we can directly use a better model rather than active learning process, to reduce the cost.

3. Also for the active learning, how to select experts is always an important question. It will be good if the authors can discuss their certria to design the expert end of active learning in their manuscript.

---

### Meta-Review · Area_Chair_fv5f · 2023-10-26

**Recommendation:** Accept (Poster)
**Confidence:** 4

**Metareview:**

In this paper, authors have proposed an information entropy-based metric for measuring data bias. To mitigate the bias, they further proposed an entropy-targeted active learning (ET-AL) framework. The proposed framework is applied to material science, which guides the acquisition of new data to improve the diversity of underrepresented crystal systems.
The paper presents an important problem of data bias mitigation and proposes novel approach to address this. Overall paper looks okay to me. However, as one of the reviewers noted, extensive evaluation of the idea against other machine learning approaches is missing.
As both reviewers lean towards acceptance of the paper, and it presents original contributions for an important problem, so I recommend acceptance.